# Persistent Dysbiosis, Parasite Rise and Growth Impairment in Aquacultured European Seabass after Oxytetracycline Treatment

**DOI:** 10.3390/microorganisms11092302

**Published:** 2023-09-13

**Authors:** Dimitris Rigas, Nikos Grivas, Aikaterini Nelli, Evangelia Gouva, Ioannis Skoufos, Konstantinos Kormas, Athina Tzora, Ilias Lagkouvardos

**Affiliations:** 1Galaxidi Marine Farm S.A., 33200 Galaxidi, Greece; rigas_dimitris@yahoo.gr (D.R.); ngrvet@gmail.com (N.G.); 2Laboratory of Animal Health, Food Hygiene and Quality, Department of Agriculture, School of Agriculture, University of Ioannina, 47100 Arta, Greece; k.nelli@uoi.gr (A.N.); egouva@uoi.gr (E.G.); jskoufos@uoi.gr (I.S.); tzora@uoi.gr (A.T.); 3Department of Ichthyology and Aquatic Environment, University of Thessaly, 38446 Volos, Greece; kkormas@uth.gr; 4Agricultural Development Institute, University Research and Innovation Centre “IASON”, Argonafton & Filellinon, 38221 Volos, Greece; 5Department of Microbiology and Microbial Pathogenesis, School of Medicine, University of Crete, 71500 Heraklion, Greece

**Keywords:** seabass, oxytetracycline, aquaculture, fish microbiome, dysbiosis, fish parasites

## Abstract

The use of antibiotics in open-water aquaculture is often unavoidable when faced with pathogens with high mortality rates. In addition, seasonal pathogen surges have become more common and more intense over the years. Apart from the apparent cost of antibiotic treatment, it has been observed that, in aquaculture practice, the surviving fish often display measurable growth impairment. To understand the role of gut microbiota on the observed growth impairment, in this study, we follow the incidence of *Photobacterium damselae* subsp. *piscicida* in a seabass commercial open-water aquaculture setting in Galaxidi (Greece). Fish around 10 months of age were fed with feed containing oxytetracycline (120 mg/kg/day) for twelve days, followed by a twelve-day withdrawal period, and another eighteen days of treatment. The fish were sampled 19 days before the start of the first treatment and one month after the end of the second treatment cycle. Sequencing of the 16S rRNA gene was used to measure changes in the gut microbiome. Overall, the gut microbiota community, even a month after treatment, was highly dysbiotic and characterized by very low alpha diversity. High abundances of alkalophilic bacteria in the post-antibiotic-treated fish indicated a rise in pH that was coupled with a significant increase in gut parasites. This study’s results indicate that oxytetracycline (OTC) treatment causes persistent dysbiosis even one month after withdrawal and provides a more suitable environment for an increase in parasites. These findings highlight the need for interventions to restore a healthy and protective gut microbiome.

## 1. Introduction

Aquaculture is the fastest-growing animal food production sector, supplying over half of the fish and seafood for human consumption [1]. Aquaculture production in Greece, for 2021, was reported to be 143,865 tons, with a value of EUR 641.96 million, which, compared to 2020, represented an increase of 8.1% in the biomass produced and an increase of 15.1% in the value of the produced product [2]. The main farmed species in Greece are the gilthead seabream (*Sparus aurata*) and the European seabass (*Dicentrachus labrax*), with production for 2021 of 125,550 tons, which represented an increase of 14.4% compared to 2020 [2]. Farmed European seabass is of key economic importance in Europe, since 96% of the total production comes from fish farming rather than fishing [3]. 

Fish gut microbiota play important roles in host health and physiology [4] by promoting development of the immune system and contributing to nutrient utilization and resistance to pathogens [5,6]. Fish gut microbial communities are also highly dynamic and respond rapidly to fluctuations in local selective pressures such as diet modification [7,8,9]. While this plasticity contributes to host adaptation to environmental changes [9,10], it can also lead to microbiota imbalances that can negatively affect fish growth and disease susceptibility [11]. Fish gut microbiota communities are influenced by a variety of factors, endogenous and external, such as host genetics, environmental factors, diet and antimicrobial compounds which have been found to affect the composition and diversity in several farmed fish, along with the emergence of resistant bacterial clones. Therefore, potential side effects on gut microbiota through antibiotic administration are a subject of interest for farmed fish, since the induced changes may involve the overall health of the fish host [12].

Among the many functions of the microbiome, it enhances the intestinal epithelial barrier, contributes to the evolution and maturing of the immune system, plays a significant role in the acquisition of nutrients that are important to the host, and above all, prevents colonization by microbial pathogens and protects against overgrowth of indigenous opportunistic microbes (pathobionts) in the case of a disturbance of the normal microbial flora [13]. In vertebrates, in the gastrointestinal tract, the microbial ecosystem that is created is neither a fixed nor a simple state but varies dynamically, contributing to both the health and nutrition of the host [14,15,16]. Most of these concepts are also applicable to fish [17]. However, fish, especially most of the farmed fish, are expected to be challenged by some of the climate change consequences due to the emergence of microbial diseases [18].

Oxytetracycline (OTC) is a broad-spectrum antibiotic that is widely used in aquaculture globally, mainly because of its effectiveness against Gram-negative and Gram-positive bacteria as well as the fact that it has minimal side effects. Oxytetracycline is a tetracycline-class antimicrobial drug that is produced by the genus *Streptomyces* of Actinobacteria. Its bacteriostatic effect is due to its ability to reversibly bind to the bacterial 30S ribosomal subunit and the microbe’s protein synthesis mechanism [19,20]. In seabass, OTC is mainly used for treatment against *Photobacterium damselae* subsp. *piscicida*, *Vibrio alginolyticus*, *Vibrio anguillarum,* and *Vibrio* spp. [21]. In seabass, an OTC daily dose must be from 75 mg/kg (recommended) to 100 mg/kg (maximum dosage) for a 10-day full treatment cycle [21].

The use of antibiotics can often shift the gut microbial community towards suboptimal compositions [22,23,24]. In many fish species, research results have indicated a total shift in the gut microbiome community combined with either similar levels of alpha diversity or a long-lasting decrement in the gut microbiome [25,26,27,28]. Dysbiosis or dysbacteriosis is defined as changes in the intestinal microbiome, either qualitatively or quantitatively, both in metabolic processes and in the concentration ratio of bacteria or their distribution in space. Despite the complexity of the relationships between microorganisms in the gastrointestinal tract, these relationships remain (except for disturbances due to adaptation or dysbiosis) stable both between microorganisms and the host and between microorganisms [29,30,31,32]. Recently, relationships of gut microbiota alteration with allergy-related, autoimmune, and metabolic diseases have been revealed [33,34]. An increase in the infiltration of bacteria and toxic substances in the gastrointestinal tract, damage to the intestinal tissues, and alterations in the functionality of digestive enzymes can be caused due to dysbiosis of the intestinal microflora, which can be caused either by improved hygiene or by using chemical additives in food [35,36,37]. In this light, the relationships between changes in the gut micro ecosystem and diseases have attracted more and more attention [38]. 

The growing need for intensive aquaculture has increased the frequency and severity of fish diseases. Given the importance of the contribution of gut microbiota to the maintenance of host health and well-being, the design of novel interventions [39] aimed at regulating the microbiota is considered to be a realistic strategy for the control of infectious diseases in fish farming [40]. Antibiotic treatment in aquaculture is necessary to keep mortality rates low, especially during bacterial infection outbreaks. However, the exact impact of antibiotics on the gut microbiome has not been well documented, leading to a new wave of research based on the impacts of antibiotics and ways to reduce them and improve fish welfare, thus preventing severe losses. However, the liberal administration of antibiotics in addition to the advantages they provide, has also led to serious problems such as the emergence and spread of resistant bacterial strains, potential food safety hazards, and major environmental issues [41]. Antimicrobial resistance has risen due to selection pressure not only on the gut microbiota of the fish but also on the environmental bacteria through horizontal gene transfer. Aquaculture could be seen as a “genetic hotspot”, ideal for gene exchange, which supplies antibiotic resistance determinants not only to natural environments but to the whole food chain [42]. This study aims to investigate the effects of OTC treatment on the intestinal microbiome in intensive seabass aquaculture.

## 2. Materials and Methods

### 2.1. Aquaculture Setup

The population of the sampled cage came from juveniles cultured in Desfina (one of the Galaxidi Marine Farm S.A. hatchery stations). On 8 March 2021, 265,000 seabass juveniles, at a weight of 3.37 g, were inserted into a round sea cage with a diameter of 60 m. Nets were changed at the suitable fish average weight, to keep the population with the highest possible water renewal. The fish were fed commercial feed based on high-quality fish meal, which had a high protein composition (46% protein and 18% fat, whereas during the first two months the feed composition was 48% protein and 15% fat). The fed was produced by Irida S.A. (Arta, Greece). Average weight sampling was performed periodically, following the Galaxidi Marine Farm’s common production policies. At mortalities exceeding 0.1%, the company’s veterinarian performed a visual inspection of the sea cage (and all cages in the same average weight category) and samplings were taken for pathogen identification. Before the first sampling, the fish were in excellent health condition, there was no pathology, and the company’s veterinarian supervised the sampling procedure.

### 2.2. Sampling

On 4 August 2021, 29 days before the OTC treatment (Figure 1), 19 random opportunistic samples of fish were collected with a scoop net from a sea cage, with an average sample weight of 45 g, 300 days post-hatching (dph). From 2 September 2021 to 13 September 2021 and from 26 September 2021 to 14 October2021, there was a mortality outbreak caused by *Photobacterium damselae* subsp. *piscicida*, with a mortality rate of 2.7% during the first period, followed by 5.6% during the second period (Figure 1). The Pasteurellosis outbreak was treated with a 120 mg/kg/day dosage of OTC, following the prescription of the company’s veterinarian, in fish feed prepared by Irida S.A. (Arta, Greece). From the same cage, on 14 November 2021, 10 samples of fish were caught, with an average sample weight of 153 g, 400 dph. All samples were immediately stored on ice and sent for sequencing within 24 h. 

### 2.3. DNA Extraction and rRNA Gene Amplicon Sequencing

The samples were sent to the “Systems Microbiology and Applied Genomics Biodetect” laboratory (Patra, Greece) for extraction and sequencing of the sample’s gastrointestinal microbiome. The fish were dissected under aseptic conditions, where the gastrointestinal tract was removed intact from each seabass. The contents of each gastrointestinal tube were extracted into a sterile phosphate solution, and the tube was rinsed with the same solution to dislodge loosely and tightly adherent microorganisms. Next, DNA was isolated from the removed material using a NucleoSpin DNA Rapidlyse Kit (Macherey-Nagel, Düren, Germany). Determination of the quantity and quality of the isolated DNA was performed using Nanodrop. Especially for purity, absorbance was measured at 260 nm and 280 nm with acceptable ratio values of 260/280~1.8.

Amplification of the 16S rRNA gene region from bacterial DNA was performed with the 341F/805R primers [43]. The amplified region of approximately 450 bases spans across the V3–V4 hypervariable regions of the 16S rRNA gene. The produced library was sequenced with Illumina MiSeq in paired-end mode (2 × 300).

### 2.4. Data Processing

Raw Fastq files were initially denoised using the UNOISE3 [44] method from USEARCH11 (64 bit version) [45], followed by clustering of Zotus with UPARSE [46] into species-level groups (97% similarity) as implemented in IMNGS [47]. The following settings were used: side trimming 10 bases, 300 < amplicon size < 500, minimum amplicon sequence variant size 4, and trim score 4. A cutoff of 0.25% was selected to remove spurious OTUs from the analysis [48]. The clusters formed were aligned and taxonomically classified with SINA [49] using the SILVA release 138 [50] as a reference. Missing classification data at the genus level and above were manually collected using EzBioCloud when possible [51]. To focus on the real intestinal microbiome, the OTUs related to cyanobacteria, chloroplasts, or mitochondria were removed from the OTU table and studied separately. A phylogenetic tree of the selected aligned sOTUs was calculated using the neighbor-joining method in MEGAX [52]. 

Downstream analysis was performed using the R scripts Rhea [53] and DivCom [54] with the default setting unless otherwise mentioned. Firstly, all taxa counts were normalized to minimum sample size (2136 reads) and into relative percentages. Afterwards, alpha diversity was measured as richness, effective richness [48], and effective Shannon. DivCom [54] was used to investigate the existence of subgroups within our dataset. Beta diversity was calculated using the generalized UniFrac method (α = 0.5) [55]. Multidimensional scale plots (MDS) and hierarchical plots were used to visualize the dissimilarity matrices. Differences between before and after OTC microbial profiles were tested with PERMANOVA (package adonis). Statistical differences in the relative abundances of taxa between groups were tested using the Kruskal–Wallis rank sum test, and the Wilcoxon rank sum test was used for pairwise comparisons. Fisher’s test was used to identify differences in the prevalences of taxa among groups. *p*-values were adjusted using the Benjamini–Hochberg procedure. Statistically significant differences were considered to be only those with an adjusted *p*-value below 5% (adjusted *p* < 0.05). Optimal growth pH for selected species was extracted from the BacDive site [56].

### 2.5. Calculation of Growth Performance

Growth performance was evaluated using the following formulas: food conversion rate (FCR) = feed intake/weight gain, specific growth rate (SGR) = 100 × [ln(final average weight) − ln(starting average weight)]/time (days), and survival ratio = (fish harvested + fish remaining in cage)/starting population. Historical data were selected from the same farm, having the same average weight at the beginning of September and feed intake and mortality ratios were measured until 14 November 2021. Two different treatments were selected as references, one treatment without the application of OTC and the other treatment with a single cycle of OTC (11 days).

## 3. Results

### 3.1. Primary Data Analysis

Sequencing resulted in a total of 1,490,206 paired-end reads (51,386 ± 11,868 reads per sample). After filtering, 1,022,085 paired-end reads (35,244 ± 10,596 reads per sample) remained, and after removing OTUs related to Eukaryota, the result was 489,898 paired-end reads (16,893 ± 7707 reads per sample) and a total of 177 bacterial OTUs. All Cyanobacteria were removed from the OTU table, except Zotu1669 from the class of Vampirivibrionia, which contained predatory bacteria of the gut and appeared only in sample F28 with 1.84% abundance.

### 3.2. Microbial Diversity Analysis and Community Profiling

Alpha diversity was measured as the effective richness and the Shannon effective metrics [48]. Before treatment, the effective richness was 28.81 ± 10.79 and the Shannon effective was 19.3 ± 7.28, whereas after treatment the effective richness was 11 ± 5.74 and the Shannon effective was 7.41 ± 4.09. The gut bacterial community post treatment had a significant reduction in alpha diversity (effective richness, *p* < 0.001 and Shannon effective, *p* < 0.001) (Figure 2a). Almost 50 OTUs contributed to the fish microbiome before OTC treatment (22 OTUs had a prevalence over 1%) and, post treatment, the number of OTUs dropped to 23 (only 10 OTUs had a prevalence over 1%). Considering beta diversity, using PERMANOVA, samples before and after antibiotics were found to be significantly different between them, forming tight clusters (Figure 2b). 

Considering average abundances of different taxa within the samples in the two sampling groups, before OTC treatment, Proteobacteria (68.79%) are the most prevalent phylum, followed by Gemmatimonadota (9.88%), Bacteroidota (7.62%), Actinobacteriota (7.08%), Firmicutes (4.05%), and Acidobacteriota (1.35%). After OTC treatment, Acidobacteriota and Gemmatimonadota are completely extinct. Proteobacteria (46.32%) are more prevalent post treatment, followed by Firmicutes (37.89%), Actinobacteriota (6.80%), Bacteriodota (4.47%), Deinococcota (3.09%), and Chloroflexi (1.41%) (Figure 2c).

At the family level (Figure 2c and Figure 3b), before treatment, common families from the Proteobacteria phylum are Caulobacteraceae (16.85%), Sphingomonadaceae (11.89%), Pseudomonadaceae (7.73%), Rhizobiaceae (5.75%), Beijerinckiaceae (3.61%), Comamonadaceae (3.34%), Xanthomonadaceae (3.31%), and Rhodobacteraceae (1.4%). From the Firmicutes phylum, common families are Bacillaceae (2.53%), Streptococcaceae (0.17%), and Staphylococcaceae (0.16%). From the Bacteroidota phylum, Cyclobacteriaceae (4.64%), Chitinophagaceae (2.46%), and Weeksellaceae (0.11%) are the most common families. Finally, from the Actinobacteriota phylum, the most common families are Microbacteriaceae (3.08%) and Micrococcaceae (1.3%). From the Acidobacteriota phylum, the most prevalent family is Blastocatellaceae (1.29%) and from the Gemmatimonadota phylum the most prevalent family is Gemmatimonadaceae (10.01%).

After treatment, from the Proteobacteria phylum, common families are Pseudomonadaceae (18.76%), Halomonadaceae (9.35%), Rhodobacteraceae (5.77%), Oxalobacteraceae (2.82%), Comamonadaceae (2.17%), Nitrincolaceae (1.68%), and Sphingomonadaceae (1.59%). From the Firmicutes phylum, common families are Bacillaceae (28.58%), Staphylococcaceae (5.84%), Streptococcaceae (5.84%), and Lactobacillaceae (1.95%). From the Bacteroidota phylum, Weeksellaceae (4.47%) is the most common family. Finally, from the Actinobacteriota phylum, the most common families are Micrococcaceae (4.56%), Corynebacteriaceae (1.15%), and Propionibacteriaceae (1.07%). From the Deinococcota phylum, the most prevalent family is Thermaceae (2.35%).

After antibiotics, the prevalences of Zotu26—*Paeobacter porticola* (99.52%), Zotu70—*Nereida ignava* (100%), Zotu87—*Kocuria palustris/assamensis* (100%), and Zotu147—*Sphingomonas changbaiensis/horti* (98.33%) increase. 

Post antibiotics (Figure 4), Zotu64—*Anaerobacillus isosaccharinicus* (97.76%) and Zotu20—*Alkalihalobacillus pseudofirmus* (100%) appear for the first time, Zotu143—*Staphylococcus epidermidis* (99.78%) and Zotu57—*Halomonas desiderata* (98.88%) increase their representation (all of them belong to the Bacillaceae family, Staphylococcaceae and Halomonadaceae), and twenty OTUs (Figure 4) that do not belong to the four families do not recover.

The increase in *Alkalihalobacillus pseudofirmus* (100%), a bacterium cultured at a pH of about 9.7, and *Halomonas desiderata*, which is also cultured at a pH between 9 and 10, indicates a significant increase in gut pH that is only harmful for fish health. Most bacteria after OTC treatment (Table 1) need a pH of over 7 (alkalic environment) to grow. 

### 3.3. Apicomplexans Presence

Sequences from OTUs classified as Cyanobacteria were further analyzed by comparison to available sequences in the NCBI. A large portion of the OTUs show the highest similarity to known cyanobacteria, but also to eukaryotes, mainly algae or apicomplexan parasites. Finding the parasites using 16S rRNA gene amplicon sequencing was possible due to the presence of conserved 16S rRNA genes in mitochondria, chloroplasts, and apicoplasts. However, the possibility of the existence of additional undetected parasites in the samples cannot be ruled out, since the primers used have low coverage of eukaryotic organelles. Two primary OTUs were identified to be of parasite origin, i.e., Zotu2 with 90.07% similarity to *Eimeria nieschulzi*, and Zotu25 with 87.94% similarity to *Cyclospora cayetanensis* apicoplasts. The very low similarities point to new, so far undescribed, apicomplexan species; however, for the sake of the analysis, we will refer to them as the *Eimeria* OTU and the *Cyclospora* OTU. After antibiotics, the proportion of reads assigned to apicomplexan parasites was significantly higher than before antibiotics (*p* < 0.001) (Figure 5a). 

Before OTC treatment, there were practically no reads assigned to the apicomplexan OTUs, while after OTC treatment, the *Eimeria* OTU appeared in most of the samples (8/10), accounting for 64.87% of the total reads, while 3/10 of the samples were positive for the *Cyclospora* OTU that was assigned 15.68% of the total reads (Figure 5b). Both apicomplexan parasites appeared after OTC treatment, possibly because of beneficial gut microbiome extinction, leading to an environment in which parasites could establish. *Eimeria* has 130 different known species that infect fish, causing coccidiosis (diarrheal disease). Against this infection, sulfonamides, ionophores, and toltrazuril are commonly used.

### 3.4. Growth under OTC Treatment

On 4 August 2021 (time point a0), the sea cage population was 215,175 fish with an average weight of 43.68 g. With feed intake of 26,135 kg until 14 November 2021 (time point a2), the sea cage population was 184,275 fish at 153.44 g. The food conversion ratio (FCR) for this period was 1.38, the special growth ratio (SGR) was 1.13, and the survival ratio was 85.64% (rather low, most of the time leading to high production cost). During August, the water temperature was mostly around 27 °C, and during September, the water temperature was around 25 °C for the first half of the month it was around 24 °C for the second half of the month. October began at 24 °C and ended at 20 °C, and then stabilized until the end of this period. The outburst of Pasteurellosis mortality ended at 22 °C. From the historical data (same average weight at the beginning and sampling at the end of the same period, grown at the same region), the cage with only one cycle (11 days same dosage) until 14 November 2021 was at 158 g (+3.3%) and the production parameters were FCR = 1.36 (−1.35%), SGR = 1.32 (+16.58%), and survival ratio = 98.2% (+14.67%), and the cage without treatment was at 179 g (+17.17%) and the production parameters were FCR = 1.31 (−4.9%), SGR = 1.39 (+22.73%), and survival ratio = 97.17% (+13.47%). 

## 4. Discussion

In our study, we investigated the effects of OTC treatment in seabass, using 16S rRNA sequencing to specify the differences between pre- (19 samples) and post-OTC (10 samples) treatments. Following the OTC treatment, as expected, gut microbiome diversity had a huge drop, causing dysbiosis. There was a significant drop in Proteobacteria and Gemmatimonas, and a significant rise in Firmicutes. Also, after OTC treatment, bacteria not detected in the samples before became a considerable part of the microbial communities. The community replacement showed the extremely delicate balance in the gut microbiome that could be disturbed when there was a drop off in diversity, with antagonistic, opportunistic, and pathogenic bacteria replacing bacteria that were useful before the treatment disturbance. The bacteria were growing in an alkaline environment (having the lowest growth pH and optimal growth pH > 7, e.g., *Alkalihalobacillus pseudofirmus* with optimal pH = 9.7 and lowest pH = 9) [56], which indicated a disturbance in gut pH. A shift in gut pH from acid to alkalic is related to slow growth ratios and disturbance in the epithelium [57,58]. Parasites of the genus *Eimeria* had a significant presence after treatment, causing either coccidiosis or a disturbance in the microbiome recovery procedure. 

The reported studies on the effects of oxytetracycline in the gut microbiome of zebrafish have shown that a low concentration of OTC did not significantly impact gut microbiome diversity, although it led to a shift in the gut community, having a significant increase in pathogenic bacteria [25]. Also, there was an increase in the variety of genes related to antibiotic resistance in the gut microbiome. Similarly, in rainbow trout, using OTC treatment at low dosages has been shown to lead to a shift in the gut microbiome [12]. This shift was caused by a decrease in Gram-negative Proteobacteria, *Bacillus*, and *Clostridium_sensu_stricto_1* and an increase in *Aeromonas*, *Brevinema*, and *Deefgea*. Also, the gut community had not stabilized until the end of the experiment (14 days after withdrawal from treatment). The decrease in *Bacillus* and *Clostridium* (both belonging to Firmicutes phylum) could also affect fish health because both are related to the enhancement of resistance to bacterial pathogens. In Nile Tilapia, after a maximum dose of OTC (100 mg/kg/day), there was no significant difference in fish growth, although the fish under treatment had less weight than the control fish [26]. Other research has had similar results, resulting in a reduction in pathways related to protein and enzyme activity causing feed inefficiency and reduced ability to digest nutrients [27,28]. Although there was no significant difference in gut microbiome diversity, there was a significant decrease in Gram-negative Proteobacteria OTUs and Gram-positive Actinobacteria OTUs. An increase in the presence of *Plesiomonas*, a Gram-negative bacterium of the Enterobacteriaceae family which is related to bacterial disease outbreaks [59], indicates that opportunistic potential pathogens that have a degree of resistance in OTC may increase their presence, mainly because of a decrease in their antagonistic bacteria due to OTC treatment. *Plesiomonas shigelloides* also increased its presence after OTC treatment in discus fish and zebrafish [60,61]. Also, in the gut microbiome of Atlantic salmon [62], there was a dominance of *Aeromonas salmonicida* after OTC treatment, a bacterium etiologically related to furunculosis.

In yellowtail kingfish, skin and gut microbiomes were sampled during and after 18 days of treatment with a combination of OTC, erythromycin, and metronidazole [63]. Also, there was fecal microbiota transplantation to a group, to measure the enhancement of gut microbiome recovery. In skin microbiome, despite the disturbance in the first three days of treatment, after the fifth day of withdrawal there was a shift to normal microbiome. There was an immediate decrease in gut microbiota diversity, and this diversity did not recover even 18 days after withdrawal from treatment. The fecal microbiota transplantation had a mixed result, having a subgroup maintain a few transplanted taxa (even from day two post FMT), but they were withdrawn by day eight post FMT. In another study, skin and gill samples from healthy and diseased (with Photobacterium damselae) European seabass were collected through a period of treatment with OTC (35 g/kg/day for eight days) and a recovery period that lasted three weeks [64]. Dysbiosis due to infection was evident in the skin rather than the gill and the skin microbiome during recovery was similar to the healthy ones.

Usage of OTC in medicated feed can lead to residuals in the local marine environment either from low absorption or uneaten medicated feed [65]. It has been estimated that 1900 kg of OTC is released in the Mediterranean Sea per year [65]. The residuals can potentially lead to antimicrobial resistance (AMR), although global warming is also associated with a rise in AMR [66]. Although the exact quantity of medicaments released in aquatic environments is not known, having a potential increase in antibiotics due to climate change emphasizes the importance of prebiotics, probiotics, and vaccines as economically viable alternatives [67]. OTC in aquaculture is used for pathogens that operate between 22 °C and 27 °C (mostly midsummer) and the predicted half-life is less than 25 min [68]. In a recent study related to chickens, Eimeria infection led to an increase in pH, partly due to decreased production of short-chain fatty acid (SCFA) [69].

Shifts in the microbiome community of seabass have been observed as a reaction to several conditions, like pollutants, feed composition or additives, and environmental factors [70,71,72,73,74]. A 2020 study by Kokou reported on the resilience of the enteric microbiome after treatment with different mixes of antibiotics [75], which was in stark contrast to our findings. Nevertheless, in the Kokou study, florfenicol was used instead of OTC, and fish were tested in closed basins with brackish water. In the field application investigated here, two rounds of OTC were used in an open-system aquaculture. A possible explanation could be that a low dosage and short applications of OTC and antibiotics, in general, can be tolerated by the fish and the microbiome will recover especially in closed systems. In open marine systems, the presence of pathogens dictates the usage of high dosages of antibiotics often in multiple rounds of treatment. The prolonged treatments open a wide window for non-symbiotic bacteria in seawater to infiltrate and establish in the fish gut before native bacteria recovery. Therefore, the natural capacity of the microbiome to recover after perturbation is related to the size of the perturbation. 

OTC treatment in European seabass showed a significant decrement in gut microbiome diversity, followed by the appearance of *Anaerobacillus isosaccharinicus* (97.76%) and *Alkalihalobacillus pseudofirmus* (100%), perhaps competing with some of the extinct bacteria. Having a long time gap from OTC treatment withdrawal (30 days) and a bad recovery of the gut microbiome, it is necessary to implement methods to accelerate the recovery curve. Also, after OTC treatment, seabass underperformed in terms of growth ratios, and it may be helpful (and profitable) for Mediterranean aquaculture to decrease this harmful impact. Fecal microbiome transplantation in yellowtail kingfish is a great example that demonstrrates there can be fast microbiome restoration, but there must be a sustainable and repeatable method. Also, the presence of parasites is only a negative factor, and they should be added in future methodologies. The rise in bacteria that need a high alkalic environment indicates at least a temporary increase in gut pH, which is related to fish growth underperformance and epithelium permeability [57,58]. Also, this environment could be beneficial for the *Eimeria* parasites, resulting in further damage to the gut equilibrium mechanism and fish health. Therefore, the mechanism of raising gut pH should be further explored.

## 5. Conclusions

In our study, after OTC treatment, there was a significant drop in alpha diversity and the appearance of bacteria that need an alkaline environment to grow. Also, after OTC, there was appearance of Apicomplexan parasites. There was no time series sampling in this study to clarify which of the three events occurred first (diversity reduction, increase in alkaline bacteria, or an increase in Apicomplexan parasites), but all of them are bibliographically characterized as harmful and each of them can be related etiologically with the growth impairment that was established from historical data. The increase in days with optimal water temperature for bacterial infections (around 22–27 °C) due to climate change will force aquaculture in longer treatment periods and even more antibiotics will be used to suppress mortality rates. Nevertheless, as we have shown, prolonged antibiotic treatments can have a cumulatively destructive effect leading to reduced fish performance and a higher economic burden.

Further research should be conducted on effective and economically viable methods for gut microbiome recovery and on alternative ways to raise fish immunity against pathogens that are related to high mortality rates (such as prebiotics, probiotics, and vaccines). This would lead to a reduced need for the extensive use of antibiotics and a reduced risk of the looming pathogen immunity for the highly used treatments. Breaking out of this vicious cycle of weaker fish, stronger infections, and heavier treatments seems to be the only sustainable way for future fish aquaculture.

## Figures and Tables

**Figure 1 microorganisms-11-02302-f001:**
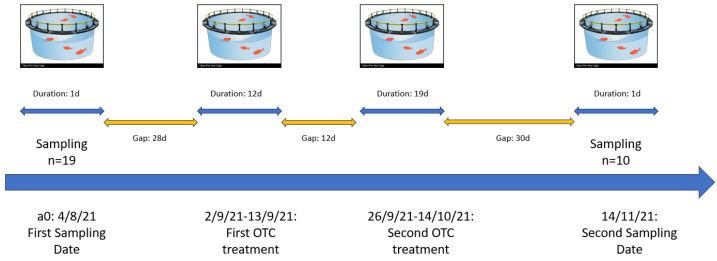
Experimental timeline. Two sampling time points (19 samples 29 days before treatment and 10 samples 30 days after the second cycle withdrawal). Two OTC cycles, one cycle of 12 days duration and another cycle of 19 days duration. Same OTC dosage at 120 mg/kg/day in medicated feed. OTC treatment was given due to an outbreak of Pasteurellosis.

**Figure 2 microorganisms-11-02302-f002:**
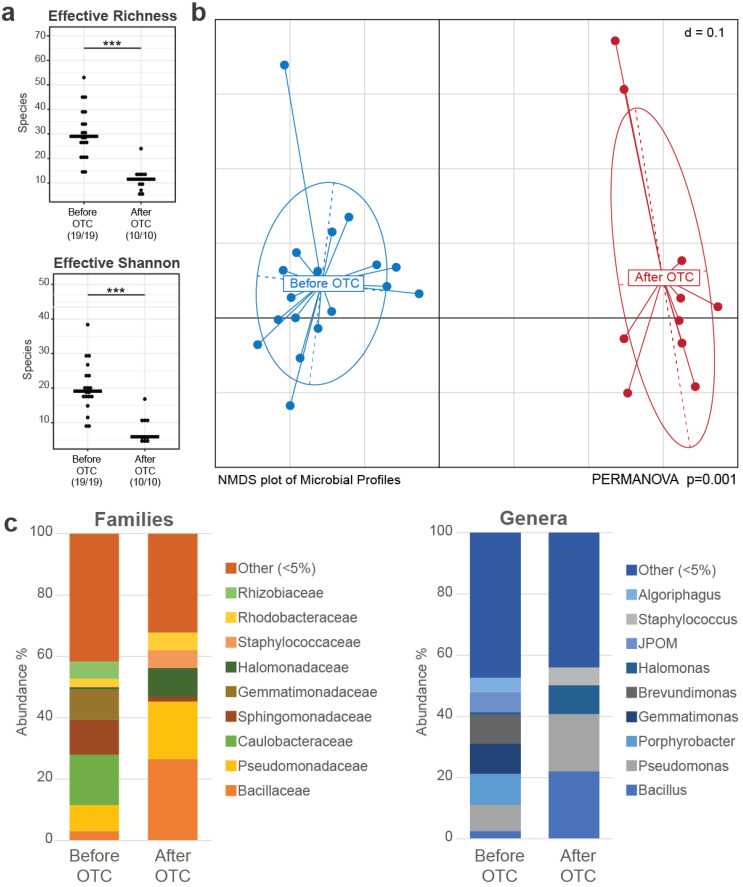
Microbial diversity of seabass samples before and after OTC treatment: (**a**) Alpha diversity in samples before and after OTC treatment. After antibiotic treatment, there is a drop in effective richness and Shannon effective measurements. Stars (above the black lines linking the two groups with a significant difference) indicate differences in number of species using the Wilcoxon rank sum test. *** adjusted *p*-value <0.001; (**b**) NMDS plot showing beta diversity among the microbial profiles from gut samples before and after OTC treatment. The two groups show significant differences in their composition (PERMANOVA *p* = 0.001); (**c**) mean profiles of samples taken before and after OTC at the family and genus levels. Differences due to treatment at the family level include an increase in Bacillaceae, Pseudomonadaceae, Halomonadaceae, Staphylococcaceae, and Rhodobacteraceae, while the families of Caulobacteraceae, Sphingomonadaceae, Gemmatimodaceae, and Rhizobiaceae showed a large drop. At the genus level, after antibiotics, the prevalences of *Bacillus*, *Pseudomonas,* and *Halomonas* increased, while the prevalences of *Porphyrobacter*, *Gemmatimonas,* and *Brevundimonas* decreased.

**Figure 3 microorganisms-11-02302-f003:**
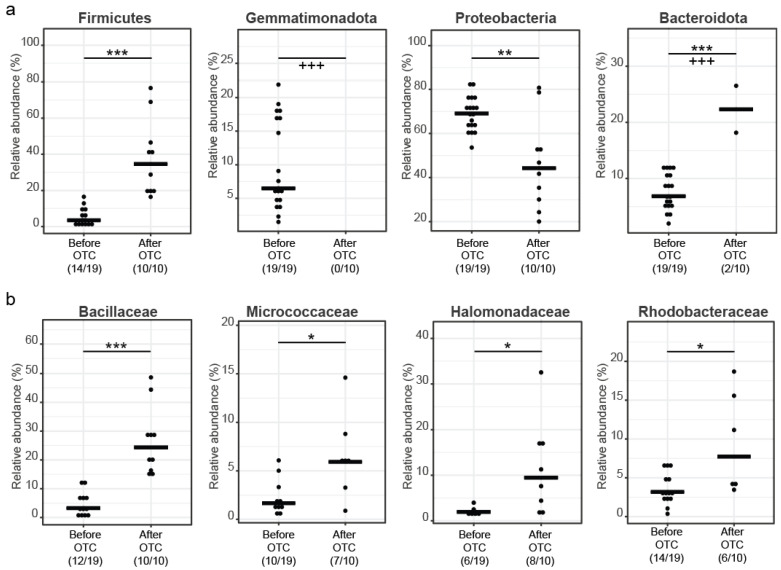
Selected statistically significant differences before and after antibiotics: (**a**) Differences at the phylum level; (**b**) differences at the family level. Statistics: Stars (above the black lines linking the two groups with a significant difference) indicate differences in relative abundance using the Wilcoxon rank sum test. Crosses (below the lines) indicate differences in prevalence, i.e., the percentage of samples positive for the given taxon, as indicated below the respective *x*-axis and tested using Fisher’s exact test. *, adjusted *p*-value <0.05; **, <0.01; *** <0.001.

**Figure 4 microorganisms-11-02302-f004:**
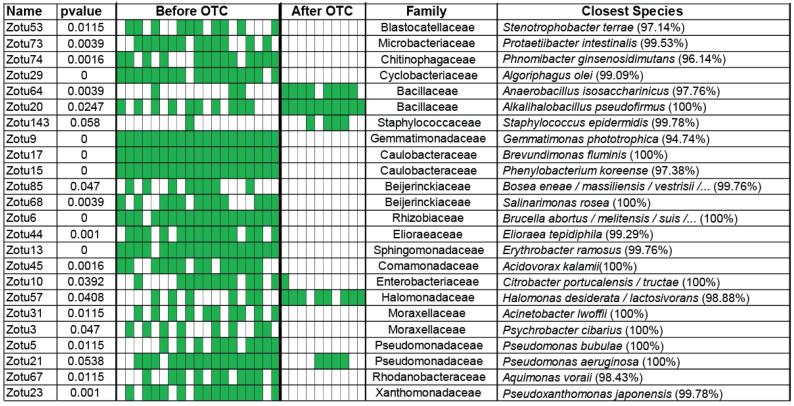
OTUs that had statistically significant differences in presence–absence data (Fisher test), indicating OTUs that were either extinguished from antibiotics or OTUs that grew after antibiotics.

**Figure 5 microorganisms-11-02302-f005:**
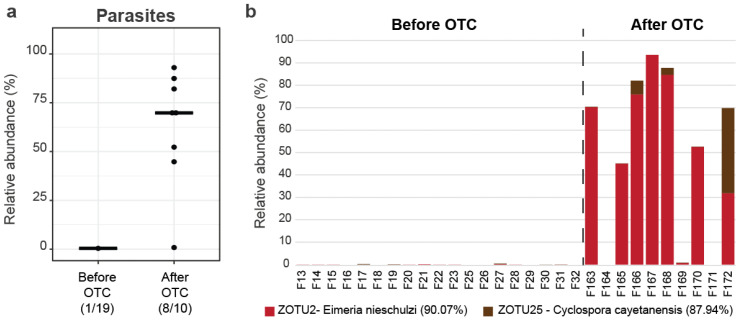
Parasitic load of fish intestinal samples before and after OTC treatment: (**a**) The relative cumulative abundance of all reads classified to be of parasitic origin across the group of samples before and after OTC treatment, Wilcoxon rank sum test *p* < 0.001; (**b**) distribution of the Apicomplexans OTUs across the samples, with their individual abundance in each sample. Both major apicomplexan OTUs (*Eimeria* and *Cyclospora*) were only detectable after antibiotics, indicating a function through which OTC application may be linked to parasitic infiltration.

**Table 1 microorganisms-11-02302-t001:** Gut bacteria after OTC treatment. Many OTUs exhibit high similarity with known species that have a high growth optimal pH [56].

OTU ID	Closest Known Species	Growth Optimal pH	Prevalence	Mean Abundance
Zotu20	*Alkalihalobacillus pseudofirmus* (100%)	9.7	10	20.9
Zotu64	*Anaerobacillus isosaccharinicus* (97.76%)	9.7	8	5.7
Zotu57	*Halomonas desiderata* (98.88)	9.5	8	11.7
Zotu26	*Phaeobacter porticola* (99.52%)	7.6	4	10.7
Zotu60	*Neptuniibacter pectenicola* (99.55%)	7.6	3	5.6
Zotu70	*Nereida ignava* (100%)	7.6	3	4.8
Zotu143	*Staphylococcus epidermidis* (99.78%)	7.3	4	14.6
Zotu86	*Pseudomonas azotoformans* (100%)	7.3	4	10.3
Zotu152	*Massilia jejuensis* (99.32%)	7.2	3	6.2
Zotu21	*Pseudomonas aeruginosa* (100%)	7	4	30.9
Zotu87	*Kocuria palustris* (100%)	7	7	6.5

## Data Availability

Original data were submitted to ENA under project accession number PRJEB64664.

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
