# Peer review of "Persistent Dysbiosis, Parasite Rise and Growth Impairment in Aquacultured European Seabass after Oxytetracycline Treatment"

_microorganisms, 2023, doi:10.3390/microorganisms11092302_

Round 1

Reviewer 1 Report

Aquaculture is one of the fastest-growing industries to fulfil the protein demand of the increasing world population, and farmed Sea Bass is of key economic importance in Europe. However, the use of antibiotics in aquaculture is sparingly regulated. In this study, the authors investigated the effects of Oxytetracycline (OTC) treatment on the intestinal microbiome in Sea Bass aquaculture following Pasteurellosis outbreak, and reported persistent gut microbiome dysbiosis, a rise of parasites and growth impairment in in Sea Bass following disease and antibiotic treatment. Overall this manuscript is well written and data nicely presented, but a few issues listed below need to be addressed.

1.      In [Introduction] the authors stated “In Sea Bass OTC daily dose must be from 75mg/kg (recommended) to 100 mg/kg (maximum dosage) for 10 days full treatment cycle [21].” But in this current study, sea bass was treated with a 120 mg/kg/day dosage of OTC for 12 / 18 days. Could it be overdose of antibiotics? It is impressive that a second Pasteurellosis outbreak happened merely 12 days after the first OTC treatment---does that indicate presence of tetracycline resistance before the initiation of the treatment?

2.      Section 2.1: according to description and figure 1, a0 occurred 28 days before OTC treatment, not 19 days.

3.      Impaired growth could be caused by depressed fish appetite due to disease, instead of the gut microbiome dysbiosis after OTC treatment. The relative weight gain rate, instead of the absolute body mass, should be a better indicator of fish growth after recovery.

4.      Considering the poor bioavailability of OTC in sea bass, it would be important if the authors can discuss the biodegradation rate of OTC in Mediterranean euryhaline environment and the potential impact of OTC residue in water/sediments on the marine environment.

Typo: [Abstract]: the “raise” of parasites should be “rise”

[Figure 2] legend: “prevelace” should be “prevalence”

[Figure 4] legend: second “either” should be “or”

Typo: [Abstract]: the “raise” of parasites should be “rise”

[Figure 2] legend: “prevelace” should be “prevalence”

[Figure 4] legend: second “either” should be “or”

Author Response

  1. In [Introduction] the authors stated “In Sea Bass OTC daily dose must be from 75mg/kg (recommended) to 100 mg/kg (maximum dosage) for 10 days full treatment cycle [21].” But in this current study, sea bass was treated with a 120 mg/kg/day dosage of OTC for 12 / 18 days. Could it be overdose of antibiotics? It is impressive that a second Pasteurellosis outbreak happened merely 12 days after the first OTC treatment---does that indicate presence of tetracycline resistance before the initiation of the treatment?

We thank the first reviewer for this comment. Although the eventual acquisition of OTC resistance for the Photobacterium damselae is a considerable risk, it has nothing to do with the two consecutive rounds of OTC treatment. Pausterellosis may need even 3 cycles to be effectively stopped and such prolonged metaphylactic treatments will eventually become a norm in aquaculture practice if warmer summers become a reoccurring phenomenon. As soon as the pathogen enters the waters it remains as a threat to young fish until the water temperature drops below 22 degrees. Therefore, since OTC treatment does not eliminate Photobacterium from the water as long as the circumstances favor Pausterellosis a new round of OTC will be applied. Pausterellosis can be rather persistent leading to high mortality rates for even 3 months (and mortality rates that overcome 50% of the population). Concerning overdose, in commercial aquaculture, medicated feed comes in multiples of 25kg (weight of a sack). So, it is often preferred to have a slight overdose in the first days (that reduces daily as the weight of fish increases), while keeping a standard feed intake every day. This is mainly because it is easier for the feeders to stay focused on a certain number of kilos per cage for a week rather than changing daily feed to keep up with a 100 mg/kg/day dose. Also, unfortunately for aquaculture, fish with Pausterellosis won’t eat at all and eventually they will die. Antibiotic treatment is not intended for the sick fish. Medicated feed is useful for post-preservation of the healthy ones or those not heavily infected. Also, the recommended dose of 100 mg/kg/day is proven insufficient in open sea systems that’s why other companies use up to 200 mg/kg/day. We modified the text to highlight the point that OTC and routine antibiotic treatments in general are a lost fight in the long run.

  1. Section 2.1: according to description and figure 1, a0 occurred 28 days before OTC treatment, not 19 days.

Corrected the text to 28 as it should be. Thanks for catching this.

  1. Impaired growth could be caused by depressed fish appetite due to disease, instead of the gut microbiome dysbiosis after OTC treatment. The relative weight gain rate, instead of the absolute body mass, should be a better indicator of fish growth after recovery.

This is a key comment since if appetite was indeed the case it would completely alter the way the results are interpreted. First, we want to make clear that sick fish do not eat and eventually die. Nevertheless, OTC makes the feed bitter so it can be assumed that healthy and infected but not sick fish dislike it and reduce their consumption eventually leading to lower weight gain. We pulled all relevant records on feed consumption for the cages in the study and compared them with historical records of the same period to calculate feed conversion rates over a month following one or no rounds of OTC.  The results show a clear reduction in feed conversion rate after one cycle of OTC, becoming more pronounced if followed by a second round like in our case. The feed consumption across those treatments was the same. Therefore, we are confident to assign the underperformance to energy loss. We altered the manuscript in several places to reflect those findings.

  1. Considering the poor bioavailability of OTC in sea bass, it would be important if the authors can discuss the biodegradation rate of OTC in Mediterranean euryhaline environment and the potential impact of OTC residue in water/sediments on the marine environment.

We thank the reviewer for this comment as it picks on a critical issue as the diffusion of antibiotics in the environment. We add in the introduction and the discussion several sentences that address the issue. In general, only a small part of OTC in the feed is absorbed and the rest is released in the water with the feces. In the surface saline water, the light can quickly remove OTC. Nevertheless, for the sinking portion of OTC, we could only refer to potential effects rather than measured effects. The sad truth is that literature is lacking in situ sediment measures for residual antibiotics in the regions of intensive aquaculture and comparisons on the resistances of isolates across those regions. We think that this is an excellent idea for a research project.

Typo: [Abstract]: the “raise” of parasites should be “rise”

Corrected

[Figure 2] legend: “prevelace” should be “prevalence”

Corrected

[Figure 4] legend: second “either” should be “or”

Corrected

Reviewer 2 Report

The manuscript "Oxytetracycline treatment led to persistent gut microbiome dysbiosis, the rise of parasites and growth impairment of European Sea Bass in open sea aquaculture" by D. Rigas et al. is devoted to the study of fish microbiome. The authors analyzed microbiome of sea bass before and after duplicated oxytetracycline treatment. The results indicated the dysbiosis in antibiotic-treated fish as well as increase of alcalophilic species and some eucariotic parazites (like Eimeria sp.). The work contains interesting observations and could be accepted after serious revision.

1. There is no results for untreated control samples. Without the comparable results it's hard to accept the presented conclusions - the shift in microbiome could be (partially) linked with the normal development/maturation. I guess the authors should provide additional data for, at least, healthy fish with same weight and age.

2. The microbiome analysis is not supported by any other data: biochemical, morphological, histological. The authors' analysis will be more argumented if it will be supported by other independent methods.

3. Minor corrections:

Discussion, p.10: Bacillus and Clostridium_sensu_stricto_1 - Please, formatting this part;

Also, in the gut microbiome of Atlantic salmon, there was a dominance of Aeromonas salmonicida after OTC treatment, a bacterium etiologically related to furunculosis. - Please, add the relevant reference for this statement.

Author Response

We want to thank the second reviewer for his time and effort in reading and evaluating our manuscript. Unfortunately, we became aware of the comments when submitting our responses to the 3 other reviewers. We do not have the time to respond appropriately in the time frame given.

Nevertheless, we want to provide some responses to the valuable comments.

This is a case study, where the application of microbiome profiling was used as a diagnostic tool in a commercial setting. The company had yearly issues with Pasteurellosis outbreaks and decided to investigate the issue. There was no control over the company practices and no special treatment of cages for the sake of this analysis. Samples were taken when fish were planned to be measured anyway. Therefore, what may look like a methodological shortcoming is simply the company protocol of operation. We need to make clear that as soon as Pastereurellosis outbreak emerges all cages with young seabass go under treatment. So, no relevant cage was left untreated for the sake of the analysis (It could have resulted in extreme mortalities).  There is no way of adding samples or other measures now. What we can do and we did is to collect historical data of other cages in past years that house fish in the same developmental stage for the same time of the year to compare growth rates. With this analysis became clear that energy extraction and utilization from food were impaired, as our fish with the same amount of food and the same temperatures gain less weight at the same time. This is an independent observation that when coupled with the microbiological data allows some hypothesis. We are very careful in the manuscript to avoid overinterpretation of the results. The facts are that 1 month after withdrawal from OTC the microbiome of the fish looks devastated. You do not need a control to evaluate that. No healthy fish microbiome is dominated by extreme alkalophiles. Since we miss dense temporal data and additional datasets we do not know the sequence of events. Did the OTC clear the normal flora and leave space for pathogenic bacteria to establish and then parasites come? Did OTC remove bacteria crucial for the defence against parasites, which colonize and lead to alkalization of the gastric system, in turn favoring alkalophilic bacteria? We do not know. Those are great questions that emerge from this study. We believe that by sharing our results that OTC treatment has a cumulative negative effect in commercial open-water seabass aquaculture we can inspire more controlled experiments and also raise awareness over alternative treatments. We encourage you to read the revised manuscript from this perspective and then evaluate the manuscript again. 

Reviewer 3 Report

The title must be revised for English improvement “ to persistent gut microbiome dysbiosis, the rise of parasites and” and also the whole manuscript need native proof reading.

The MS without line number.

Sea Bass could be written as “seabass” along the MS.

Food could be written as “feed” along the MS.

Revise the abbreviation along the MS for example OTC in the abstract must be mentioned in full first to abbreviated form.

The sentence needs to rewrite to be more clearer “In addition, antibiotics administered in the body could disturb the healthy gut microbial community and cause an imbalance of microbiota called dysbiosis [22–24].”

The sentence needs to rewrite to be more clearer “Knowing that in aquaculture practice usage of antibiotics can lead to a growth reduction, microbial profiling before and after the treatment can offer a better understanding of the possible role microbes play in this observation.”

In the M & M you have to descript the studied species and farming condition in details first before mentioning sampling subtitle.

The OTC treatment regime could have a Ref.

The determination of growth performance was conducted by a primary method not accepted method. It could be recalculated according to the initial and final weight of the specimen and could include weight gain, specific growth rate, and also the feed intake and feed conversion ratio must be mentioned. This could be mentioned in M&M and results as a separate subtitles.

Could you correct Fig 1 were the period between the first sampling and first OTC treatment mentioned as 28d but it could 19 days according to the description.

In Fig. Caption, Don’t start sentence with number “2 OTC cycles,”. Also, the numbers less than ten could be written in words.

In first paragraph of the discussion you mentioned the results of parasites before the description of bacteria, could you reorder this information or delay the part of Parasite to the second half of the discussion

This part need Refs “Those bacteria are growing in an alkaline environment (having lowest growth pH and optimal growth pH>7, e.g. Alkalihalobacillus pseudofirmus with optimal pH=9.7 and lowest pH=9), indicating a disturbance in gut pH. Shift in gut pH from acid to alkalic, is related to slow growth ratios and disturbance in the epithelium.”

The discussion mis to provide a clear interpretation of the effect of OTC on the marked increase of the parasite community in seabass gut after treatment. This is interesting and must be discussed in a separate section supported with recent published studies.

“After treatment, there was a presence (for the first time) of gut bacteria that need high pH to grow,” this not the first report for this information.

The conclusion is wrong constructed, it like discussion, the good conclusion must written based on the obtained results and presence a take home message.

The title must be revised for English improvement “ to persistent gut microbiome dysbiosis, the rise of parasites and” and also the whole manuscript need native proof reading.

Author Response

We are grateful to the third reviewer for his time and comments. We believe that through his detailed instructions, the manuscript has been clearly improved.

Below are the comment-by-comment responses.

The title must be revised for English improvement “ to persistent gut microbiome dysbiosis, the rise of parasites and” and also the whole manuscript need native proof reading.

The title has been shortened and revised to” Persistent dysbiosis, parasite rise and growth impairment in aquacultured European Seabass after oxytetracycline treatment.”. All the text went over careful proofreading.

The MS without line number.

We agree with the reviewer that line numbers are helpful for the review process. Nevertheless, we use the publisher's template and we are not clear if adding line numbers is allowed. We will contact the publisher to ask about this issue.

Sea Bass could be written as “seabass” along the MS.

We replace Sea Bass with seabass across the manuscript

Food could be written as “feed” along the MS.

We replace food with feed across the manuscript

Revise the abbreviation along the MS for example OTC in the abstract must be mentioned in full first to abbreviated form.

We follow the reviewer's recommendation and use the full form of all abbreviations on their first use.

The sentence needs to rewrite to be more clearer “In addition, antibiotics administered in the body could disturb the healthy gut microbial community and cause an imbalance of microbiota called dysbiosis [22–24].”

Modified

The sentence needs to rewrite to be more clearer “Knowing that in aquaculture practice usage of antibiotics can lead to a growth reduction, microbial profiling before and after the treatment can offer a better understanding of the possible role microbes play in this observation.”

Removed

In the M & M you have to descript the studied species and farming condition in details first before mentioning sampling subtitle.

A section on fish farming conditions has been added in the manuscript M&M

The OTC treatment regime could have a Ref.

The recommended dosage of OTC had a reference. As we now highlight in the manuscript, the dosage used in practice is a decision of the company's veterinarian.

The determination of growth performance was conducted by a primary method not accepted method. It could be recalculated according to the initial and final weight of the specimen and could include weight gain, specific growth rate, and also the feed intake and feed conversion ratio must be mentioned. This could be mentioned in M&M and results as a separate subtitles.

This is a very good recommendation. We followed the suggestion of the reviewer and now we have a much better description and quantification of the observed growth impairment. We thank again the reviewer for his proposal.

Could you correct Fig 1 were the period between the first sampling and first OTC treatment mentioned as 28d but it could 19 days according to the description.

Corrected

In Fig. Caption, Don’t start sentence with number “2 OTC cycles,”. Also, the numbers less than ten could be written in words.

Corrected across the manuscript.

In first paragraph of the discussion you mentioned the results of parasites before the description of bacteria, could you reorder this information or delay the part of Parasite to the second half of the discussion

We agree with the reviewer that the order was not natural. We restructured the discussion as per reviewer recommendation.

This part need Refs “Those bacteria are growing in an alkaline environment (having lowest growth pH and optimal growth pH>7, e.g. Alkalihalobacillus pseudofirmus with optimal pH=9.7 and lowest pH=9), indicating a disturbance in gut pH. Shift in gut pH from acid to alkalic, is related to slow growth ratios and disturbance in the epithelium.”

References were added.

The discussion mis to provide a clear interpretation of the effect of OTC on the marked increase of the parasite community in seabass gut after treatment. This is interesting and must be discussed in a separate section supported with recent published studies.

“After treatment, there was a presence (for the first time) of gut bacteria that need high pH to grow,” this not the first report for this information.

We guess this is a misunderstanding. We did not claim that those bacteria were first reported in our analysis. We rather point out that alkalophilic bacteria were not there before the treatment and appeared after treatment. We modify the sentence to avoid confusion.

The conclusion is wrong constructed, it like discussion, the good conclusion must written based on the obtained results and presence a take home message.

We rewrite the conclusion to highlight the point that antibiotic treatment comes with a direct and an indirect cost. Direct when counting the cost of the treatment and indirect from the side effects of the treatment to the host and the environment. We hope our findings will motivate scientists to investigate alternative solutions to general antibiotics.

Thanks again for accepting the task of reviewing and for your valuable comments.

Reviewer 4 Report

The manuscript approaches a worthy of investigation topic. Besides, it is well-written and presents numerous relevant findings. No issues regarding methodological aspects were identified, and the results are sound with the objectives proposed. I have a few suggestions to improve the manuscript:

1) The title is too long and must focus on the study's main findings. It could be enhanced;

2) The introduction is too long and must be shortened to at least 30% of the word count. Furthermore, the data described in the first paragraph should be actualized. Please use more recent references;

3) The conclusion section requires a detailed revision and should focus on an objective summary of the findings and their potential match with the proposed objectives.

Author Response

We like to express our gratitude to reviewer 4 for helping with the review process. Below are our responses to the expressed comments and suggestions.

  • The title is too long and must focus on the study's main findings. It could be enhanced;

The title has been shortened and revised to:

” Persistent dysbiosis, parasite rise and growth impairment in aquacultured European Seabass after oxytetracycline treatment.”.

  • The introduction is too long and must be shortened to at least 30% of the word count. Furthermore, the data described in the first paragraph should be actualized. Please use more recent references;

We removed much of the introduction and more recent references were used as suggested.

  • The conclusion section requires a detailed revision and should focus on an objective summary of the findings and their potential match with the proposed objectives.

We radically changed the conclusion. We hope that our message on the self-limiting nature of general antibiotics and the need for alternative solutions in aquaculture is now clear.

Thanks again for the suggestions. Shortening some parts really helps readability. We had to add some parts according to other reviewers' suggestions but overall we believe that the manuscript has been significantly improved.

Round 2

Reviewer 2 Report

Thanks for authors' explanations, but I hope to see more detailed analysis of microbiome shift based on other literature data (if we could not obtain the relevant control). For example:

https://doi.org/10.3390/jmse11051022

https://doi.org/10.1016/j.scitotenv.2021.150402

https://doi.org/10.3389/fmicb.2022.831034

https://doi.org/10.1007/s12602-022-09974-w

https://doi.org/10.1038/s41598-020-78441-9

It's just the first results from the "european sea bass microbiome" request in google.scholar. The topic is studying very intensive. In the current discussion the authors focused on the antibiotic treatment effect for different fish species, but what is relevant for the sea bass aquaculture in general?

It's interesting what is the role of the main bacterial taxa presented in the "normal" microbiome? Whether the microbiome shift affects some biochemical and morphological characteristics?

Minor corrections:

Title: Please, delete the dot from the end of the article title;

Keywords: Replace Sea Bass to Seabass for uniform style;

Discussion, p.11: Bacillus and Clostridium_sensu_stricto_1 - Please, formatting this part;

Discussion, p.11: Also, in the gut microbiome of Atlantic salmon, there was a dominance of Aeromonas salmonicida after OTC treatment, a bacterium etiologically related to furunculosis. - Please, add the relevant reference for this statement.

Author Response

Many thanks for your understanding of the limitations of our study and also for sharing the relevant literature. Especially the Kokou study was an embarrassing miss from our side.  We devote one paragraph to the discussion of their findings and the reflection on our discoveries.

We did not extend much on the discussion of what is normal microbiome. This is a very big question and needs a combined effort among researchers to be resolved. We still do not know what constitutes a healthy microbiome even for humans.  

All syntactical errors or typos noted are now fixed.

Thanks again

Reviewer 3 Report

I would like to appreciate the authors efforts in improving the MS according the suggestion of reviewers. just minnor correction nedded.

in M and M aquaculture setup.

3.37gr correct to 3.37 g (the abbreviation of gram is g) consider along the MS.

descripe the used feed ( it is a commercial diet) so mentioned the composition protein and fat %m, the producing company, place and country.

in growth calculation: it was  in a wrong way it could be as follows:

Feed conversion ratio (FCR)=feed intake/weigh gain

Specific Growth Rate ( SGR)=100 × (Ln final weight - Ln starting weight)/time (days).

the language still need minor improvement especially the new inserted sections

the language still need minor improvement especially the new inserted sections

Author Response

Thanks again for your valuable contribution to our work.

We went through all of your recommendations and even found a few extra.

We hope our findings raise awareness of the possible negative effects of OTC on the microbiome and encourage new research on possible alternatives.
